# Stochastic step-wise feature selection for Exponential Random Graph Models (ERGMs)

Helal El-Zaatari[1]*, Fei Yu[2], Michael R. Kosorok[1]

**1** Department of Biostatistics, University of North Carolina, Chapel Hill, NC, United States of America,
**2** Health Sciences Library, University of North Carolina, Chapel Hill, NC, United States of America

* helal@unc.edu

## Abstract

This study introduces a novel methodology for endogenous variable selection in Exponential Random Graph Models (ERGMs) to enhance the analysis of social networks across various scientific disciplines. Addressing critical challenges such as ERGM degeneracy and computational complexity, our method integrates a systematic step-wise feature selection process. This approach effectively manages the intractable normalizing constants characteristic of ERGMs, ensuring the generation of accurate and non-degenerate network models. An empirical application to nine real-life binary networks demonstrates the method's effectiveness in accommodating network dependencies and providing meaningful insights into complex network interactions. Particularly notable is the adaptability of this methodology to both directed and undirected networks, overcoming the limitations of traditional ERGMs in capturing realistic network structures. The findings contribute to network analysis, offering a robust framework for modeling and interpreting social networks and laying a foundation for future advancements in statistical network analysis techniques.

**Data Availability Statement:** All relevant data are within the paper and its Supporting information files.

## 1 Introduction

Statistical analysis of social networks plays a pivotal role in various scientific disciplines, offering valuable insights into complex network interactions. Accurate modeling is particularly crucial when working with moderately sized networks, typically comprising a few thousand nodes, as it enables the explanation, analysis, replication, and prediction of network phenomena observed in nature. In the field of health sciences, social network analysis contributes to reducing health disparities [1] and fostering collaboration and research efficiency, leading to scientific innovations and discoveries [2]. By uncovering patterns in collaboration networks, network analysis facilitates the prediction of future connections among individuals or organizations, which holds significant value for multiple stakeholders including health policy researchers, administrators, and research sponsors [3–5].

The advancement in computational power of personal computers in the 21st century has empowered researchers to conduct sophisticated statistical modeling without relying on supercomputers [6]. One powerful technique widely used in social network research is Exponential Random Graph Models (ERGMs). ERGMs are particularly adept at capturing network

**Funding:** The author(s) received no specific funding for this work.

**Competing interests:** The authors have declared that no competing interests exist.

dependencies by incorporating endogenous variables. However, a challenge arises when the chosen endogenous variables do not accurately capture the observed network structures, leading to ERGM degeneracy [7], a state where networks become unrealistic and uninterpretable [7–9].

Addressing the weakness of ERGMs presents multiple challenges that require careful consideration. First, the dependency among network observations, similar to that in longitudinal studies, invalidates models that assume independence [10]. Secondly, accurately modeling this dependency is crucial but complicated. While stochastic block models treat this dependency as a nuisance parameter, ERGMs aim to explicitly model and quantify it through endogenous variables. The complexity lies in selecting appropriate endogenous variables, given the vast array of choices and the risk of degeneracy through inappropriate selections. Researchers have reported at least five distinct types of ERGMs in their scholarly work, including the standard ERGM [11, 12], Bayesian ERGM [13], Temporal ERGM [14], Separable Temporal ERGM [15, 16], and Multi-level ERGM [17, 18]. They face the daunting task of choosing from thousands of potential endogenous variables [19], a process lacking systematic guidance or tools.

Therefore, this study proposes and tests a novel methodology for endogenous variable selection in ERGMs, targeting the critical challenges in the field. Our approach encompasses key aspects of variable selection, degeneracy screening, and model fitting, providing a comprehensive solution to enhance the effectiveness and reliability of ERGM modeling, particularly in collaboration networks [20]. We conduct empirical testing and rigorous analysis for the proposed statistical algorithms, aiming to facilitate more accurate and meaningful interpretations of network phenomena in various scientific disciplines.

The paper is structured as follows: Section 2 offers a mathematical definition of an ERGM, setting the foundational understanding of this class of models. Section 3 delves into the details of the endogenous variable selection procedure, which includes establishing the initial set of variables, a novel step-wise variable selection process, and an innovative degeneracy screening method based on edge count. Section 4 applies our proposed algorithms to nine real-life binary networks, presenting numerical results that include the selection of endogenous variables, the count of potential pairwise ERGMs, average counts of edges, 2-stars, and triangles, and the efficacy of our degeneracy screening approach. Section 5 discusses the significance of our methodology and testing results, potential future research directions, and limitations, underscoring the implications of our findings for advancing the field of network analysis.

## 2 Definition of ERGMs

A network or graph $G$ consists of nodes and edges denoted by $G = (V, E)$ respectively. The nodes are assumed to be finite with $V = \{1, \ldots, N\}$. The edges represent ties between two different nodes $i, j$. Modeling networks is centered around the edges $E$ of a graph. The outcome of interest $Y_{i,j}$ is defined for two separate nodes $i \in V$ and $j \in V$. Depending on the type of network, this outcome $Y_{i,j}$ can take on binary, discrete or real valued numbers. For example, a binary outcome where $Y_{i,j} = 1$ indicates an edge between nodes $i$ and $j$ while $Y_{i,j} = 0$ indicates no edge. Additionally, nodes $i \in V$ can possess a collection of attributes situated in Euclidean space.

Statistical modeling of a network involves defining a probability distribution over graph G. This model space comprises a set of these probability distributions, each indexed by a parameter space $\Theta$. The selected probability distribution will determine the complexity and details of the network model. Inspired by generalized linear models, exponential random graphs model the probability of a tie formation $Y_{i,j} = 1$ given the nodal attributes $X$. For a binary outcome, ERGMs are akin to logistic regression in network data analysis [21]. Analogous relationships

exist between ERGMs with discrete and continuous-valued ties and their generalized linear model counterparts, such as Poisson regression and Gamma regression, respectively. The formulation for an ERGM with a binary outcome is given below.

$$P_\theta(Y = \boldsymbol{y}|\boldsymbol{\theta},\boldsymbol{x}) = \psi(\boldsymbol{\theta}_1,\boldsymbol{\theta}_2)exp\{\boldsymbol{\theta}_1^T s(\boldsymbol{y}) + \boldsymbol{\theta}_2^T g(\boldsymbol{y},\boldsymbol{x})\} \tag{1}$$

The network with $n$ nodes is represented via an adjacency matrix $Y$ on support $\mathcal{Y}_n$ and $y \in \mathcal{Y}_n$ [22]. The vector of endogenous variables is represented via $s(\boldsymbol{y})$ where $s : \mathcal{Y}_n \to \mathbb{R}^p$ and the vector of exogenous variables is represented via $g(\boldsymbol{y})$ where $g : \mathcal{Y}_n \to \mathbb{R}^q$. The natural numbers $p, q \in \mathbb{N}$ depend on the number of endogenous and exogenous variable chosen. The vector of regression coefficients, $\boldsymbol{\theta}$, is partitioned into two sub-vectors $\boldsymbol{\theta}_1$ and $\boldsymbol{\theta}_2$. The nodal attributes are represented by $\mathbf{x}$. The computationally intensive normalizing constant is $\psi(\boldsymbol{\theta}_1,\boldsymbol{\theta}_2)$, and a vector of exogenous variables, $g(\boldsymbol{y},\boldsymbol{x})$, with their associated regression coefficients represented by $\boldsymbol{\theta}_1$ and $\boldsymbol{\theta}_2$ respectively.

## 3 Endogenous variable selection for ERGMs

In this section, we introduce a novel step-wise feature selection methodology for ERGMs, which adapts the classical statistical method of forward selection technique [23] to address the challenges posed by the complexity of selecting endogenous variables for ERGMs. Our approach starts by focusing on ERGMs with two predictors and employs the Akaike Information Criterion(AIC) for model assessment. This methodological framework guides the initial selection of endogenous variables, their subsequent evaluation, and categorization based on AIC impacts, and concludes with advanced degeneracy screening techniques.

### 3.1 Obtaining an initial set of endogenous variable

The selection of endogenous variables for an ERGM presents a significant challenge due to the vast array of available options, including hundreds of pre-defined or user-customized variables. These variables are integral for modeling different network structures [24]. In this study, we start with thirteen commonly used endogenous variables, as identified from the ERGM package in R [21]. These variables, selected for their relevance to binary un-directed networks include kstar, degree-wise shared partners (dsp), non-edgewise shared partners (nsp), edge-wise shared partners (esp), triangle, isolates, sociality, degree cross product, degree popularity, geometrically weighted edgewise shared partners (gwesp), geometrically weighted non-edge-wise shared partners (gwnsp), geometrically weighted dyad-wise shared partners (gwdsp) and geometrically weighted degree. The kstar, esp, dsp and nsp endogenous variables require an upper bound whereas the other nine endogenous variables do not require an upper bound. These endogenous variables represent characteristics of social networks such as reciprocity, transitivity and centrality. For example, the triangle term measures the transitivity of a network. If a triangle term is included in an ERGM with positive regression coefficient it means that transitivity is a key feature in the observed network [25].

We employ a systematic method to select endogenous variables by establishing an informed upper bound, thus providing a structured way to refine choices, particularly for variables requiring a natural number input. For example, the dsp variable is a network statistic equal to the number of dyads with k shared partners [19], and demands the selection of an input $k \in \mathbb{N}$. Given the vast range of possibilities, it's necessary to set an upper limit for k. Similarly, variables such as kstar, esp, and nsp also require a natural number to be well-defined [21]. Different natural numbers k lead to distinct network structures, as illustrated by the difference between dsp(2) dsp(3) (Fig 1).

**Fig 1. Illustration of the dyadwise shared partner endogenous variable with _k_ = 1, _k_ = 2 and _k_ = 3 respectively.**
(Source: The original graph appeared as Fig 11 in [26]).

Addressing the impracticality of considering an infinite yet countable number of endogenous variables, our approach involves sequentially fitting uni-variate ERGMs. Starting from $k = 1$ and progressing until we reach a specific cutoff point, $k = N_k$. This cutoff, $N_k$ is determined when we achieve three or more consecutive parameter estimates reaching infinity. Beyond $N_k$, further consideration of endogenous variables becomes impractical, as they lead to coefficient estimates of negative infinity. The identification of $N_k$ plays a pivotal role in dictating the size of the initial set of endogenous variables. Applying this method to an observed network results in a set of M candidate endogenous variables, $\mathcal{S}_M$, with the set's size determined by the upper bounds of variables like dsp, esp, nsp, kstar and the remaining nine endogenous variables.

## 3.2 Stochastic forward selection

This section delves into the Stochastic Forward Selection process, the core of our proposed methodology. We start with a basic ERGM featuring only an edge term, akin to the intercept in a linear model. This baseline model, also known as a Bernoulli Random Graph assumes that the probability of a tie formation between two nodes follows a Bernoulli distribution independent of other ties within the network [10]. This assumption does not account for observation dependence, an important factor in network data.

The process involves the following steps, detailed in Algorithm 1 (Bounding the input Parameter k) and Algorithm 2 (Stochastic Forward Selection for Endogenous Variables).

**Algorithm 1** Bounding the Input Parameter $k$

1. **Endogenous Variable Requirement:** Start with an endogenous variable `s_i(`**`y`**`, l)` with an input parameter `l` $\in \mathbb{N}$
2. **Sequential Fitting:** Fit uni-variate ERGMs with an edge term and endogenous variable `s_i(`**`y`**`, l)` beginning with `l = 2`.
3. **Upper Bound Determination:** Identify the upper bound number $N_k$ when the parameter estimates for `s_i(`**`y`**`, N_{k+2})`, `s_i(`**`y`**`, N_{k+1})`, `s_i(`**`y`**`, N_k)` all yield negative infinity.
4. **Variable Iteration:** Repeat 1–3 for the following endogenous variables: dsp, esp, nsp and kstar, marking their respective upper bounds by $N_1$, $N_2$, $N_3$ and $N_4$.
5. **Final Set Formation:** Obtain a final set of endogenous variables $\mathcal{S}_M$.

Building upon the Algorithm 1, we categorize the endogenous variables $s_i(y) \in \mathcal{S}_M$ based on their observed relative AIC changes during the stochastic forward selection process. This categorization is essential in discerning variables that significantly enhance the model and those that may lead to degenerate models or yield ambiguous results.

- Category 1: Endogenous variables that consistently lower the AIC compared to the null ERGM, indicating a positive contribution and suggesting their inclusion in the model.

- Category 2: Variables that lead to degenerate ERGMs, marked by a very negative relative AIC change or a consistently negative mean relative AIC change, suggesting their exclusion.

- Category 3: Variables with ambiguous impact on the AIC, possibly due to poor initial parameter estimates or lack of predictive power. Here, the 10th percentile of the relative AIC change, denoted as $\hat{b}_i$, is crucial. A positive $\hat{b}_i$ indicates potential predictive power, while a negative value suggests exclusion.

Therefore, the null model serves as a baseline for comparing candidate ERGMs. We systematically fit uni-variate ERGMs for each element in s $s_i(y) \in \mathcal{S}_M$ for $i \in \{1, \ldots, M\}$ as shown in Eq (2).

$$P_\theta(Y = \boldsymbol{y}|\boldsymbol{\theta}) = \psi(\theta_0, \theta_i)exp\{\theta_0 s_0(\boldsymbol{y}) + \theta_i s_i(\boldsymbol{y})\} \qquad i \in \{1, \ldots, M\} \tag{2}$$

The edge term and its associated regression coefficient are represented by $s_0(\boldsymbol{y})$ and $\theta_0$, respectively. The endogenous variable under consideration is $s_i(\boldsymbol{y})$, with its regression coefficient $\theta_i \in \mathbb{R}$. The null ERGM's AIC is denoted by $AIC_0$, and the AIC for each uni-variate ERGM is by $AIC_i$ for $i \in \{1, \ldots, M\}$. The selection of the endogenous variable, $s_i(\boldsymbol{y})$ is contingent upon the calculated relative AIC change as shown in Eq (3).

$$b_i = \frac{AIC_0 - AIC_i}{AIC_0} \quad i \in \{1, \ldots, M\} \tag{3}$$

**Algorithm 2** Stochastic Forward Selection for Endogenous Variables
1. **Initial Set Formation:** Start with a set $\mathcal{S}_M = \{s_1(y), \ldots, s_M(y)\}$ consisting of $M$ candidate endogenous variables.
2. **Null Model Fitting:** Fit a null model ERGM with only an edge term, denoting the AIC value by $AIC_0$.
3. **Variable Assessment:** For $i \in \{1, \ldots, M\}$:
   - Sequentially fit uni-variate ERGMs with one endogenous variable at a time; $P_\theta(Y = \boldsymbol{y}) = \psi(\theta_0, \theta_i)exp\{\theta_0 s_0(\boldsymbol{y}) + \theta_i s_i(\boldsymbol{y})\}$.
   - Record the estimate of the AIC for each uni-variate ERGM by $AIC_i$ and compute the relative AIC change $b_i$.
   - Refit the uni-variate ERGM $M$ times and compute the 10th percentile of the relative AIC change denoted by $\hat{b}_i$.
4. **Variable Exclusion:** If $\hat{b}_i \leq 0$ then remove $s_i(\boldsymbol{y})$ from the set $\mathcal{S}_M$.

This forward selection strategy effectively categorizes endogenous variables based on their influence on the AIC. Variables that consistently elevate the AIC are considered less informative and excluded. The 10th percentile of the relative AIC change, $\hat{b}_i$, plays a crucial role in this process. If $\hat{b}_i$ is positive, it suggests that this endogenous variable can predict network structure; otherwise, it should not be included. The AIC is computed via MCMC with starting values obtained via contrastive divergence [8].

## 3.3 Degeneracy screening

After categorizing and selecting endogenous variables for inclusion, we now address a critical aspect of model refinement in ERGMs: degeneracy screening(Algorithm 3). This step is vital to ensure that our model remains robust and accurate, free from the distortions of multicollinearity often observed in ERGMs with multiple endogenous variables [7]. Degenerate networks, characterized by unrealistic network structures, emerge as a significant challenge in ERGMs when multicollinearity occurs due to the inclusion of numerous endogenous variables. To counteract this, our approach analyzes network motifs—small, statistically significant graph patterns typically comprising up to 6 nodes [26]. These motifs serve as a barometer for assessing the realism and practicality of the networks generated by our ERGM. To discard degenerative ERGMs, model selection needs to be based on network motif counts.

**3.3.1 Network motifs used for model selection.** The use of network motifs for model selection is an extension of the degeneracy screening, which enables us to delve deeper into the structural analysis of the networks. To recommend non-degenerate ERGMs we count the number of edges for an observed network and compare that count to the average number of edges a candidate ERGM produces. We then compute the relative error of these counts in the last step. While the edge network motif played a central role in screening for degenerate models, our model selection process employs additional network motifs to refine the selection criteria further. Specifically, we focus on the counts of 2-stars and triangles alongside the edge counts. The rationale behind incorporating these motifs is to compare the distribution of these specific patterns in the candidate ERGMs against their occurrence in the observed network.

The selection of models is based on the alignment of the mean number of edges, 2-stars, and triangles in the ERGMs with their corresponding counts in the observed network. The closer these averages are to the observed counts, the more representative and accurate the model is considered. This approach ensures that the selected ERGM not only avoids degeneracy but also closely mirrors the actual network structure in terms of these key motifs.

**Algorithm 3** Degeneracy Screening Algorithm

1. **Edge Count Assessment:** For an observed network, compute the observed edge counts denoted by $H_O$
2. **Model Comparison:** For a proposed ERGM $M \in \mathcal{M}$, compute the average edge count denoted by $H_M$.
3. **Discrepancy Calculation:** Compute the difference $|H_M - H_O|$.
4. **Model Exclusion:** If $\frac{|H_M - H_O|}{H_O} \geq 4$ then discard $M$ from the set of possible models.

The last step of the degeneracy screening algorithm is critical: if a candidate ERGM's average edge count significantly deviates from the observed count (either overestimates or underestimates), it is deemed degenerate and thus excluded from our set of potential models. This step ensures the models we select not only statistically represent the observed network structure but also adhere to realistic network formations.

By integrating degeneracy screening into our methodology, we enhance the model's reliability, ensuring that it reflects the true nature of the network data and remains free from the distortions of multicollinearity. This process, combined with our earlier steps of variable selection and categorization, culminates in an ERGM that is both robust and representative of the complex dynamics inherent in network structures.

## 4 Numerical studies

This section presents the application of our algorithms to nine real-life networks [21, 27], varying in complexity and size (i.e., the nodes vary from 16 to 418).

## 4.1 Types of networks

These networks are categorized into three main categories based on their structure and size, allowing us to comprehensively evaluate the potential and limitations of our method across varied network types. (1) Small Networks: This category includes networks with up to 20 nodes and a maximum of 30 edges. Representative networks in this group are the Florentine marriage, Florentine business and Molecule networks [28]. (2) Moderately Complex Networks: Networks in this category are slightly larger and more complex, with node counts ranging from 20 to 80 and edge counts between 50 and 200. Examples include the Lazega lawyer network, Kapferer tailor shop networks 1 and 2, and the Zach karate networks [29, 30]. (3) Highly Complex Networks: the final category encompasses the largest and most complicated

network with nodes ranging from 80 to 418 and edges counts from 200 to 556 [31]. A notable network in this category is the challenging Ecoli network [32], known for its complexity.

## 4.2 Stochastic forward selection(Algorithm 1 and 2)

In applying Algorithm 1, we obtained an initial set of endogenous variables, establishing a model space for each of the 9 networks (Table 1). The upper bounds of kstar, nsp, dsp and esp are found in Table 2.

The model space consists of ERGMs with either one or two distinct endogenous variables alongside the edge term. Aiming to select endogenous variables that accurately reflect the observed network structure, we applied Algorithm 2, leading to a significant reduction in the model space for all networks. An example of such an ERGM, utilizing the kstar(2) and kstar(3) terms is visualized Fig 2. This visualization represents three networks simulated from this ERGM, showcasing Algorithm 2's effectiveness in capturing network dynamics.

Transitioning from the broader application of our proposed stochastic forward selection approach above, the following is a specific example highlighting the importance of the relative AIC percentile in our algorithm. We applied algorithm 1 to a transcription regulation network for Ecoli [32–34], a case that exemplifies the nuances of variable selection in Category 3. In this instance, the relative AIC change between AIC change between the null ERGM and the uni-variate ERGM was recorded 90 times. The focal endogenous variable was the degree cross product. The results, depicted in Fig 3, show that in 5 out of 90 instances, there was a substantial increase in AIC compared to the null model. Crucially, the 10th percentile of the relative

**Table 1. Characteristics of 9 real-life undirected networks obtained from the literature.**

| Network: | # of Nodes: | # of Observed Edges: | # of 2-stars | # of Triangles: |
|---|---|---|---|---|
| Lazega | 36 | 115 | 1852 | 240 |
| Kapferer | 39 | 158 | 3132 | 402 |
| Kapferer2 | 43 | 190 | 4074 | 504 |
| Zach | 34 | 78 | 1056 | 90 |
| Molecule | 20 | 28 | 120 | 12 |
| Faux Mesa High School | 205 | 203 | 1318 | 124 |
| Ecoli | 418 | 519 | 10580 | 84 |
| Florentine Marriage | 16 | 20 | 94 | 6 |
| Florentine Business | 16 | 15 | 72 | 10 |

**Table 2. The upper bounds of kstar, esp, dsp and nsp for the 9 real life networks.**

| Network: | Upper Bound of kstar | Upper Bound of esp | Upper Bound of nsp | Upper Bound of dsp |
|---|---|---|---|---|
| Lazega | 14 | 6 | 7 | 7 |
| Kapferer | 23 | 11 | 7 | 11 |
| Kapferer2 | 24 | 12 | 7 | 12 |
| Zach | 16 | 9 | 5 | 9 |
| Molecule | 4 | 2 | 2 | 2 |
| Faux Mesa High school | 12 | 4 | 3 | 4 |
| Ecoli | 71 | 6 | 9 | 9 |
| Florentine Marriage | 5 | 2 | 2 | 2 |
| Florentine Business | 4 | 2 | 2 | 2 |

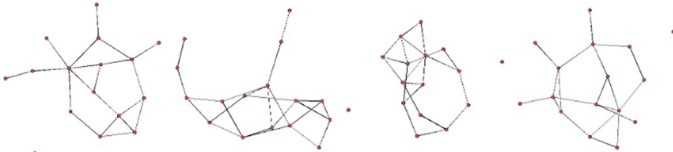

**Fig 2. Leftmost network denotes the observed Florentine marriage network.** The remaining three networks are draws from the ERGM in equation $P_\theta(Y = y|X) = \psi(\theta)exp\{\theta_0 \times edges + \theta_1 \times kstar(2) + \theta_2 \times kstar(3) + \theta_3 g_1(x) + \theta_4 g_2(x) + \theta_5 g_3(x)\}$. $g_1(x)$ denotes the exogenous variable of familial wealth, $g_2(x)$ denotes the number of seats on the civic council and $g_3(x)$ denotes the total number of business and marriage ties.

AIC for the degree cross product is positive with a median value of 0.0265. This observation underscores the potential risk of incorrectly excluding significant variables based on a single AIC calculation, thus validating the need for a multi-faceted evaluation approach as embodied in our methodology.

## 4.3 Degeneracy screening and model selection (Algorithm 3)

Our focus of model selection was primarily on the counts of edges, 2-stars and triangles as determined by Algorithm 3. This approach allowed us to systematically identify and exclude degenerate ERGMs, which are characterized by edge counts that significantly diverge from those observed in the actual networks.

For this reduced model space, the average number of edges, 2-stars and triangles generally aligned with the observed counts. However, the number of edges tended to be overestimated across all networks, a bias often inherent in exponential family models [35]. To address this discrepancy, it is advisable to allow ERGMs to include more than 3 endogenous variables, thereby enhancing model accuracy. The variations in the counts of 2-stars and triangles compared to the observed networks further illustrate the complex dynamics within these networks and the necessity of a flexible and robust modeling approach.

For this reduced model space, the average number of edges, 2-stars and triangles tend towards the observed count. The number of edges were overestimated for all networks as shown in Table 3. This bias is inherent to exponential family models [35] and can be corrected if more endogenous variables are used. On the other hand, the average number of 2-stars and triangles were sometimes above the observed count and below the observed count for different

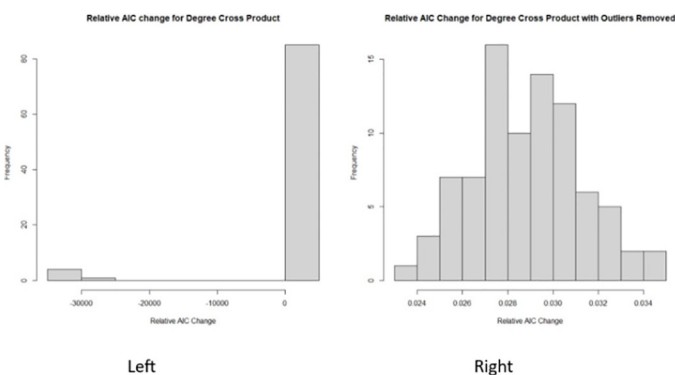

**Fig 3. Relative AIC change frequency.** Frequency of the relative AIC change for the uni-variate ERGM with degree cross product as the main predictor. The histogram on the left is an example of relative AIC fluctuation due to poor initial starting points. The histogram on the right exhibits relative AIC fluctuation that mimics random noise.

**Table 3. Results from applying our algorithms to the 9 real-life networks.**

| Network: | # of Models | Average # of Edges: | Average # of 2-stars | Average # of Triangles: |
|---|---|---|---|---|
| Lazega | 85 | 224.03 | 1924 | 174.2 |
| Kapferer | 128 | 288.8 | 2387 | 207.14 |
| Kapferer2 | 150 | 293.2 | 2792 | 222 |
| Zach | 16 | 117.6 | 1061 | 81.3 |
| Molecule | 69 | 29.68 | 123.4 | 3.98 |
| Faux Mesa High School | 39 | 344.4 | 754.3 | 96.90 |
| Ecoli | 104 | 869 | 2057 | 9.54 |
| Florentine Marriage | 1 | 24.97 | 147.5 | 12.59 |
| Florentine Business | 2 | 22.80 | 59.13 | 23.93 |

Note: counts of the average network motifs generated by the ERGMs proposed by our methods are depicted in columns 2 to 4.

networks. A remedy to this discrepancy is allowing the candidate ERGM to possess more than 3 endogenous variables.

## 5 Discussion

The results of the numerical studies have affirmed the effectiveness of our proposed methodology in generating well-fitting and non-degenerate ERGMs. Our approach has been successfully applied to a diverse range of networks, from small-scale networks with fewer than 20 nodes to complex networks like the Ecoli network, encompassing both directed and undirected structures. This versatility in application demonstrates the robustness of this approach.

The novelty of our method lies in its adaptive use of a step-wise feature selection process tailored specifically for ERGMs. This approach, which meticulously evaluates the impact of each endogenous variable using the AIC, marks a significant departure from traditional methods. It addressed the inherent complexity of ERGMs, especially the challenges posed by their intractable normalizing constants, and offers a robust framework that enhances the model's accuracy and interpretability.

Our methodological framework opens new avenues in network analysis, particularly in handling directed networks, which traditionally pose a challenge due to their complex structures. The ability to incorporate directed endogenous variables, though resulting in an expanded model space, paves the way for more nuanced analyses of directed networks. This capability is crucial for future studies to understand the directional dynamics within networks, offering potential for groundbreaking discoveries in fields such as social network analysis, epidemiology, and beyond. One notable limitation of our approach is the inflated model space that emerges when incorporating directed endogenous variables. This expansion complicates the model selection process, particularly when models include more than three variables. Addressing this limitation necessitates the development of sophisticated model selection procedures capable of navigating this increased complexity.

## 6 Conclusion

In conclusion, the method we introduced in this study represents a significant advancement in network analysis. By providing a robust and adaptable framework for ERGMs, our methodology not only ensures the generation of accurate and non-degenerate models but also enhances the potential for their application in more complex network types. While the challenge of an expanded model space presents an opportunity for further research, it also highlights the fertile

ground for further advancements in the analytical capabilities and applications of ERGMs. As we build on this work, the potential for new insights and understandings of complex network structures becomes increasingly attainable.

## Supporting information

**S1 Data. Contains the edge lists and vertex attributes for the nine networks used in this study as excel files.** Contains the R code for the three algorithms and their results in a .RData file.
(ZIP)

## Author Contributions

**Conceptualization:** Helal El-Zaatari, Fei Yu, Michael R. Kosorok.

**Data curation:** Helal El-Zaatari.

**Formal analysis:** Helal El-Zaatari, Michael R. Kosorok.

**Investigation:** Helal El-Zaatari, Fei Yu, Michael R. Kosorok.

**Methodology:** Helal El-Zaatari, Michael R. Kosorok.

**Project administration:** Helal El-Zaatari.

**Resources:** Helal El-Zaatari, Fei Yu.

**Software:** Helal El-Zaatari.

**Supervision:** Fei Yu, Michael R. Kosorok.

**Validation:** Helal El-Zaatari, Fei Yu, Michael R. Kosorok.

**Visualization:** Helal El-Zaatari, Fei Yu.

**Writing – original draft:** Helal El-Zaatari.

**Writing – review & editing:** Helal El-Zaatari, Fei Yu.

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
