## [Decision Letter · Decision Letter 0]

10 Jul 2024

PONE-D-24-10857Stochastic Stepwise Selection for Exponential Random Graph Models (ERGMs)PLOS ONE

Dear Dr. El-Zaatari,

Thank you for submitting your manuscript to PLOS ONE. After careful consideration, we feel that it has merit but does not fully meet PLOS ONE’s publication criteria as it currently stands. Therefore, we invite you to submit a revised version of the manuscript that addresses the points raised during the review process.

Both reviews recommend you should revise your manuscript. Please make the suggested changes. I agree that the paper is interesting and has potential for publication after a major revision. Please make the suggested changes to better highlight your work. If you can address the reviewer's concerns, provide a point-by-point response with your revision. Show all changes in the manuscript text file with track changes or color highlighting. If you can't address specific requests or find points invalid, explain why in the point-by-point response.

We look forward to receiving your revised manuscript.

Kind regards,

Pablo Martin Rodriguez

Academic Editor

PLOS ONE

Journal Requirements:

3. In the online submission form, you indicated that "Commonly used R packages for Exponential Random Graph models contain all the original data. The results of this manuscript will be provided by the first author."

5. Please amend either the title on the online submission form (via Edit Submission) or the title in the manuscript so that they are identical.

6. Please amend the manuscript submission data (via Edit Submission) to include author  Dr. Fei Yu and Dr. Micahel R. Kosorok.

7. Please ensure that you refer to Figure 2 in your text as, if accepted, production will need this reference to link the reader to the figure.

Reviewers' comments:

Reviewer's Responses to Questions

**Comments to the Author**

1. Is the manuscript technically sound, and do the data support the conclusions?

Reviewer #1: Partly

Reviewer #2: Yes

2. Has the statistical analysis been performed appropriately and rigorously? 

Reviewer #1: No

Reviewer #2: Yes

3. Have the authors made all data underlying the findings in their manuscript fully available?

Reviewer #1: Yes

Reviewer #2: Yes

4. Is the manuscript presented in an intelligible fashion and written in standard English?

Reviewer #1: Yes

Reviewer #2: Yes

5. Review Comments to the Author

Reviewer #1: Please see the attached referee report.

Reviewer #2: The manuscript is well developed and its subject is scientifically relevant and challenging. The proposed method was applied in extensive datasets (networks) and had good performance in selecting endogenous variables in ERGMs. Some questions to be addressed are in the attached document.

6. PLOS authors have the option to publish the peer review history of their article (what does this mean?). If published, this will include your full peer review and any attached files.

Reviewer #1: No

Reviewer #2: No

---

## [Author Response · Author response to Decision Letter 0]

31 Aug 2024

Dear Editors and reviewers,

Thank you for reviewing our manuscript, Stochastic Stepwise Feature Selection for Exponential Random Graph Models. We appreciate the comments from the reviewers. Following this letter are the reviewers’ original comments with our responses, including how and where the manuscript was modified. The revision (marked in red) has been developed in consultation with the co-authors, and each author has given approval to the final form of this revision.

Sincerely,

Helal El-Zaatari

---

## [Decision Letter · Decision Letter 1]

29 Oct 2024

PONE-D-24-10857R1Stochastic Step-wise Feature Selection for Exponential Random Graph Models (ERGMs)PLOS ONE

Dear Dr. El-Zaatari,

Thank you for submitting your manuscript to PLOS ONE. After careful consideration, we feel that it has merit but does not fully meet PLOS ONE’s publication criteria as it currently stands. Therefore, we invite you to submit a revised version of the manuscript that addresses the points raised during the review process.

You will see that one of the reviewers recommends that you revise your manuscript. Note that this is a minor revision, so please consider making the suggested changes.

We look forward to receiving your revised manuscript.

Kind regards,

Pablo Martin Rodriguez

Academic Editor

PLOS ONE

Journal Requirements:

Reviewers' comments:

Reviewer's Responses to Questions

**Comments to the Author**

1. If the authors have adequately addressed your comments raised in a previous round of review and you feel that this manuscript is now acceptable for publication, you may indicate that here to bypass the “Comments to the Author” section, enter your conflict of interest statement in the “Confidential to Editor” section, and submit your "Accept" recommendation.

Reviewer #1: All comments have been addressed

Reviewer #2: All comments have been addressed

2. Is the manuscript technically sound, and do the data support the conclusions?

Reviewer #1: Yes

Reviewer #2: (No Response)

3. Has the statistical analysis been performed appropriately and rigorously? 

Reviewer #1: Yes

Reviewer #2: (No Response)

4. Have the authors made all data underlying the findings in their manuscript fully available?

Reviewer #1: No

Reviewer #2: (No Response)

5. Is the manuscript presented in an intelligible fashion and written in standard English?

Reviewer #1: Yes

Reviewer #2: (No Response)

6. Review Comments to the Author

Reviewer #1: (No Response)

Reviewer #2: (No Response)

7. PLOS authors have the option to publish the peer review history of their article (what does this mean?). If published, this will include your full peer review and any attached files.

Reviewer #1: No

Reviewer #2: No

---

## [Author Response · Author response to Decision Letter 1]

31 Oct 2024

I have included the data and R code needed to reproduce the findings of the paper. They are found in a folder as supplementary information.

---

## [Editor Report · Decision Letter 2]

13 Nov 2024

Stochastic Step-wise Feature Selection for Exponential Random Graph Models (ERGMs)

PONE-D-24-10857R2

Dear Dr. El-Zaatari,

We’re pleased to inform you that your manuscript has been judged scientifically suitable for publication and will be formally accepted for publication once it meets all outstanding technical requirements.

Kind regards,

Pablo Martin Rodriguez

Academic Editor

PLOS ONE
---

## [Editor Report · Acceptance letter]

1 Dec 2024

PONE-D-24-10857R2 

PLOS ONE

Dear Dr. El-Zaatari, 

I'm pleased to inform you that your manuscript has been deemed suitable for publication in PLOS ONE. Congratulations! Your manuscript is now being handed over to our production team.

Kind regards, 

on behalf of

Professor Pablo Martin Rodriguez 

Academic Editor

PLOS ONE